# Research on Motivational Mechanisms and Pathways for Promoting Public Participation in Environmental Protection Behavior

**DOI:** 10.3390/ijerph20065084

**Published:** 2023-03-14

**Authors:** Weidong Chen, Kaisheng Di, Quanling Cai, Dongli Li, Caiping Liu

**Affiliations:** 1Department of Management and Economics, Tianjin University, Tianjin 300072, China; 2College of Politics and Public Administration, Qinghai Minzu University, Xining 810000, China; 3College of Chunming, Hainan University, Haikou 570228, China

**Keywords:** environmental protection, public participation, policy implementation, pathway analysis

## Abstract

Public participation in environmental protection is an essential component of evaluating the effectiveness of ecological and environmental protection. General awareness, social dynamics, and cognitive preferences frequently impact the protection’s impact. The aim of this study is to investigate the correlation research on the confluence of mainstream awareness, social factors, and cognitive preferences by building a theoretical model. First, this work employs partial least squares structural equation modelling (PLS-SEM). Second, using the mediation model, the research describes and examines the factors that motivate public involvement in ecological and environmental conservation. Third, the research summarizes the suggested path countermeasures to offer practical advice and helpful ecological and environmental protection solutions. The findings demonstrate that mainstream policy leadership substantially impacts environmental conservation. Leadership in policy matters restricts the group’s natural awareness of social factors. The subjective quality and competence basis in cognitive preferences are significantly influenced by policy leadership. Policy leadership significantly influences the effectiveness of environmental protection through the mediating factor of cognitive preferences. The ability base has a considerable mediating effect on cognitive preferences.

## 1. Introduction

The central government has been stepping up the environmental examination of local governments, and environmental protection in China has recently taken on a high-pressure stance [1]. The public’s engagement in performance evaluation is a crucial component of environmental protection assessment, a methodical effort including the government, businesses, and the general public. Enhancing the efficiency of public engagement in environmental protection performance assessment is one of them. By doing so, the adverse effects of public value conflicts on the effectiveness of environmental governance can be successfully mitigated [2].

A kind of “social environmentalism” is the public’s involvement in environmental conservation. It supports public participation in environmental protection initiatives, sees environmental protection as a civil right, and sees the general public as its wards and participants [3]. The social environment is an intricate, dynamic system. According to social environmentalism, it is essential to consider how people’s social environments affect their behavioral choices and thoughts, behaviors, and outcomes [4]. Encouraging public involvement in environmental protection can assist in increasing public knowledge and duty to participate in environmental protection activities and improve the performance outcomes of environmental protection evaluation. Although the social environment is objectively conducive to public involvement in environmental protection at this point [5], what elements influence the people’s passion for participation, and what is most important among them?

Performance refers to a system or process’s actions or outcomes [6]. Performance is a result, genuine behavior, or performance [5]. According to the theory of planned behavior, a model of behavioral explanation, organizational drivers have a significant role in determining human behavior, perceptions and attitudes, behavioral skills, environment, and behavior outcomes [7]. In light of the cascading environmental strain and condensing workload of the central government, what are the realistic ways to enhance public participation in environmental protection performance?

Due to this, some specialists have examined it from a widespread cultural standpoint. According to Susan (2016), ideological transition into policy orientation determines how people behave in society [8]. For instance, Wang Yanhong (2011) emphasizes that creating policies, particularly the goal of creating a beautiful China, improves the leadership and timeliness of policies [9]. Policy tool innovation was suggested by Li Shuzhuo et al. (2021) to increase the anticipated economic costs and ecological advantages [10]. From the standpoint of social factors, specific experts have conducted research. For instance, Chen and Shuisheng (2020) stressed the importance of explicitly cultivating multidimensional values through group engagement to increase everyone’s innate understanding of the need to conserve the environment [11]. Some professionals have researched using cognitive preferences. In Germany, for instance, individual norms have a definite positive orientation on the governance of environmental contamination situations, according to Li et al. (2022) [12]. Generally, more research has been conducted on the impact of public involvement in environmental protection due to specific influencing elements such as popular culture, social factors, and cognitive preferences. However, only a few studies have looked at how the interaction of integrated elements and the effect of environmental protection interact.

Based on the previous discussion, a structural equation model is built in this paper to examine the full impact of subjective and objective factors, including mainstream consciousness, social factors, and awareness preferences, on the efficacy of public participation in environmental protection. In addition to increasing the effectiveness of public participation in environmental protection, the paper also provides theoretical support to enhance the efficiency of government governance in environmental protection. The paper examines the practical path to promote public participation in environmental protection performance under the combined influence of multiple factors.

## 2. Theoretical Basis and Assumptions

### 2.1. Public Participation Theory and Mainstream Awareness of Environmental Protection

Ideology reflects mainstream consciousness, and macro policies, in turn, influence it [8]. China encourages citizen involvement in environmental conservation to create a beautiful China and develop pertinent policies [13]. Public involvement theory has a lengthy history. Public participation has drawn increased attention in theory and practice since 1990 [14]. The idea of governance and the practice of public participation have evolved from an early concentration on coalition networks to a change to meta-governance emphasizing coordinated governance procedures and a shift to spontaneous involvement [15]. As a crucial component of efficient risk communication and workforce planning, it emphasizes the significance of a conditional relationship between perceptions and behavior [16]. According to the notion that this process also embodies the class reality and principles of inclusive governance, the validity of the right to public participation is assured by dismantling antidemocratic groups and consolidating power in private interests [17]. As the leading organization responsible for creating policy and preserving the environment, public participation in national policy choices has proven effective [18]. In addition to how institutions influence the development and implementation of policy, the relationship between participatory path dependency and internal dynamics and interactions among public participants is expanding [5,19]. Given the growing normalization and diversification of many kinds of participation, including the Internet, it is critical to strengthen the analysis of public involvement pathways from the standpoint of the governance system [20].

Sherry Angstein first proposed the ladder theory of civic engagement in his 1969 paper, “The Ladder of Citizen Engagement” [21]. The ladder concept is always used to practice public interaction in China. The changing nature of consciousness stresses how conscious behaviour is stimulated in a specific conscious context, and the ladder theory contends that consciousness directly regulates behaviour [2]. The growth of a beautiful China was the impetus for the rise of an ecological civilization. People’s environmental protection awareness rises due to the interaction of various elements, including culture and environmental conditions [22]. Varied groups have different viewpoints on the public’s knowledge of environmental protection, whether directly impacted by their interests or driven by environmental presumptions. The public’s subjective perceptions of behaviour connected to environmental preservation and incentive to act are usually subject to sweeping changes. However, the informational connection between the environment and behaviour reduces the divergence in public policy, improves the foundation of the mainstream discourse that promotes public participation, and produces a good increase in environmental protection consciousness.

Political psychology and science academics think political ideology interacts with multidimensional advocacy and education to shape all states of consciousness. Participation is influenced by societal ideas about environmental conservation [23]. It operates on the fervent wish that government institutions take particular measures to protect their interests and direct policy orientation toward building a beautiful China [24]. People will be better equipped to comprehend scientific data and, as a result, have a greater appreciation for scientific findings the more people are influenced by advocacy and have more knowledge [25]. As a result, the method of communication and the means of communication work together to increase environmental attitudes, for instance, regarding many critical environmental issues, such as the ozone layer hole, noise pollution, and living environment issues [26]. Beautiful China’s growth promotes the development of policy support, promoting a pro-government stance and making it more straightforward for authority and participation to coexist in their respective qualities [15].

Given that environmental protection awareness predominates in participation behaviour, the public tends to deal with information barriers similarly to asymmetry through information reception, affecting participation performance overall [27]. The dimensions of action between people’s active and passive involvement commonly need to be clarified when the public participation perspective needs to be articulated. The public consciousness in a particular setting creates information channels and methods from its participatory actions and meaningful symbiosis while being influenced by outside environmental satisfaction, which forms the basis for individual cognition, judgment, and ultimately behavioral decisions [28]. In light of the analyses mentioned earlier, the following hypotheses are presented.

**Hypothesis 1** **(H1).**
*Policy support impacts public participation basis.*


### 2.2. Social Factors and Perceived Preferences for Public Participation in Environmental Protection

Social factors influence both the cognitive preferences for public involvement and the creation of spontaneous group awareness [7]. In turn, public participation in environmental protection group consciousness and competence are expressed by cognitive preferences among them [4]. Most social learning specialists agree that learning occurs at the personal and action levels. Public participation skills and the associated knowledge are described in their two most important aspects at the personal level [29]. It translates objective rules into new knowledge that pertains to the issue at hand and adapts to the area of action that supports the action’s subject through the varied interweaving of the subjective drive of consciousness and the objective cooperation of behaviour [30]. In order to modify the actual action and the action of the general public in environmental preservation, there must be a dialectical union between the subjects of any action’s consciousness and ability to act [31]. Even though consciousness frequently dictates many policy circumstances, consciousness is a precursor to behaviour. It frequently helps to improve personal consciousness and catalyzes particular environmental modifications so that participants are empowered to provide valuable information and “collectively participate in public action” [32]. Policy involvement illustrates the causal relationships between the subjective elements of the process by which information awareness is transformed into shared knowledge among a broader group. A common goal is to involve the public in problem-solving techniques or critical analysis of perspectives and skills, exchanging knowledge to reach an agreement [33]. Discussions among diverse stakeholders and information sharing among policy providers are standard components of this strategy.

At the individual level, civic engagement and the associated knowledge and skills are characterized by their two most important dimensions [29]. It changes objective laws by repeatedly weaving together the subjective drive of consciousness and the objective collaboration of behaviour to create new knowledge that pertains to the situation at hand and is tailored to the area of action that supports the action’s topic [30]. To modify the action and the action of the general public in environmental preservation, the dialectical unity between the subjects of any action’s consciousness and ability to act plays a vital role in promoting and supporting this transformation [31]. Even though consciousness often dictates many policy circumstances, it is a precursor to behaviour. It frequently aids in raising individual consciousness and sparks specific environmental alterations so that participants are empowered to address issues and “collectively participate in public action” [32]. It exemplifies how the subjective elements of the process by which information awareness is transformed into shared knowledge among a wider group are causally connected. A common goal is to include the public in problem solving or critical thinking on viewpoints and abilities [33]. Information is exchanged between parties to reach an agreement. This strategy typically entails interactions between multiple parties and information sharing among policymakers.

**Hypothesis 2a** **(H2a).**
*Policy support impacts public participation awareness.*


**Hypothesis 2b** **(H2b).**
*Policy support impacts public participation capability.*


**Hypothesis 3a** **(H3a).**
*Public participation awareness impacts public participation basis.*


**Hypothesis 3b** **(H3b).**
*Public participation capability impacts public participation capability.*


**Hypothesis 4** **(H4).**
*Public participation awareness impacts public participation capability.*


### 2.3. Dual Nature of Public Participation in Environmental Protection Performance

There are objective and subjective components to public participation in environmental protection performance [6]. The general public’s support for environmental protection policies reveals a policy act orientation in which it is crucial to consider the more excellent institutional framework [34]. Some interventions might not be as successful and supported when implemented with other policies as when employed alone. For instance, a recent study contends that the effects of the early situation are considerably supported by legislation. Of course, substantial variability between places within a single dimension may exist. Creating public policies promoting behaviour change is normative [35]. Interventions in policy behaviour may be effective for one group but ineffective for another due to the dynamic balance between objective policy support and natural performance behaviour, which has evolved into a fundamental social norm across all known cultures [36]. Reciprocity promotes rational self-interest for the vast majority of each group’s members and the collective. The importance of pressuring individuals who have the power to influence policy to stress reciprocity is usually included in comparisons between objective policy support and actual performance.

Between policy subjects and participating subjects, satisfaction and actual functioning are developed into the policy’s input base [37]. The absence of civic competence affects the use of participatory processes and public participation in strategic decision making, as revealed by over half of the respondents in [38]. The study found that the frequency of favorably adopted proposals increases with participant competence and inclusion. Public managers are more confident in the results of participatory processes they believe to be more capable. One study found that more inclusive processes and participants resulted in suggestions that local city councilors believed were appropriately informed about current policy challenges [39]. The following hypotheses were produced based on the interactions between the participant base and awareness protection.

**Hypothesis 5a** **(H5a).**
*Policy support impacts actual function of performance.*


**Hypothesis 5b** **(H5b).**
*Policy support impacts performance satisfaction.*


**Hypothesis 6a** **(H6a).**
*Public participation awareness impacts actual function of performance.*


**Hypothesis 6b** **(H6b).**
*Public participation awareness impacts performance satisfaction.*


**Hypothesis 7a** **(H7a).**
*Public participation capability impacts performance satisfaction.*


**Hypothesis 7b** **(H7b).**
*Public participation capability impacts actual function of performance.*


**Hypothesis 8a** **(H8a).**
*Public participation basis impacts actual function of performance.*


**Hypothesis 8b** **(H8b).**
*Public participation basis impacts performance satisfaction.*


We created a theoretical framework diagram for this study by combining the assumptions above with our analysis and research on the theoretical framework and by consulting certain professionals and academics, as shown in Figure 1.

## 3. Methodology

### 3.1. Questionnaire Distribution

Because collecting nationwide survey samples using precise probability sampling is challenging, this study used Questionnaire Star, a recognized online questionnaire survey platform. As the leading research site, we primarily chose the province of Qinghai, where there is good public involvement in environmental conservation. We conducted the questionnaire interviews at random locations while following the survey’s requests. The authors employed the questionnaire research platform to carry out a similarity test and an invalid questionnaire filtering review process on the data collected to assure the data quality’s reliability and accuracy. These methods resulted in the collection of 453 valid questionnaires for this study, of which 48.8% (*n* = 221) of the respondents were men, 51.2% (*n* = 232) were women, and 87.6% (*n* = 453) held a bachelor’s degree or higher. The demographic characteristics of the respondents are detailed in Table 1.

### 3.2. Measurement of Variables

In order to look into the correlations in the hypothesis, we created a questionnaire. Then, based on a thorough literature review, we created a questionnaire. We next checked for conceptual equivalency by translating it into Mandarin and then back into another form of English. In order to analyze the questionnaire, we also conducted in-depth interviews with leading academics who have made significant theoretical contributions to this area of public participation in environmental protection research. We specifically invited the professionals and academics to complete a preliminary version of the questionnaire and evaluate its accuracy and any challenges they faced while completing it. We altered how some of the questions were presented in response to the experts’ suggestions to ensure that all presentations were consistent with practice.

In the paper, specific questions for each component are provided in Table 2. In particular, we depended on questions from Han Wang et al. [40] and Cui et al. [41] for public involvement, social elements, and policy support, respectively. We used the Qiu and Hatch et al. questions created for public participation awareness preferences [42,43]. We based our questions for the public participation performance on those created by Capmourteres et al. [44]. Respondents were asked to rate their level of agreement with each construct on a 7-point Likert scale, with 1 representing “strongly disagree” and 7 representing “strongly agree”.

### 3.3. Methodology Analysis

The model outlined in this study requires the simultaneous analysis of linkages between several variables, as shown in Figure 1. In public administration research, structural equation modelling is typically employed to examine these linkages [45]. SEM is typical because it can simultaneously test several associations in a single model. Two standard techniques for employing structural equation modelling to determine causal links are covariance-based structural equation modelling (CB-SEM) and partial least squares structural equation modelling [46]. CB-SEM and PLS-SEM have been used in numerous social science investigations because of their advantages. However, each has advantages and disadvantages [47]. Statistically, PLS-SEM is a variance-based method, whereas CB-SEM is a covariance-based method.

On the other hand, the variance-based PLS-SEM approach is employed in exploratory research to create novel theories. The main application of CB-SEM is theory testing. Due to theoretical and methodological issues, PLS-SEM has gained greater traction than CB-SEM. The PLS-SEM method is excellent in explaining the variance of predictive structural relationships because it emphasizes maximizing the explained variance of endogenous latent variables more than the perfect replication of the theoretical covariance matrix. Predictive analysis using the PLS-SEM approach is instrumental when working with highly complex data. This technique estimates the latent variables through components that are precise linear combinations of the indicators assigned to the latent variables.

PLS-SEM is better suited for conventional regression regarding robustness parameters because the specified model parameters are constant in the sample [48]. The literature also asserts that PLS, which has fewer restrictions than the more widely used SEM, is more appropriate for theory construction [49]. Since this study addresses the technique of theory building in greater detail, the model in this paper is investigated using the partial least squares (PLS) method. In this study, PLS is the criterion for our estimation technique. In conclusion, PLS-SEM enables integrated models for risk assessment to get beyond the limitations of the multivariate standard distribution assumption and satisfies this paper’s prediction and explanation-oriented exploratory research objectives. Prior studies have shown that PLS-SEM offers essential benefits when handling more intricate mediated effects models. SmartPLS (version 3.0), a JAVA-based PLS-SEM program with strong platform compatibility, is employed for data processing and analysis.

## 4. Analysis and Results

### 4.1. Results of Reliability and Validity Testing

Validity evaluation of the data. The questionnaire for this study included 28 questions [50]. The 453 pieces of data from the sample were examined for internal consistency using the reliability coefficient before an exploratory component analysis was performed to ensure the scale had internal consistency. A Cronbach’s value of 0.867 indicated that the test’s outcome indicated that the data exhibited internal consistency.

To construct the research data so that they were more scientific and based on theoretical assumptions, exploratory factor analysis was performed on the data scale before the structural equation model was built. This validity analysis was used to argue that the data could be used. In the validation results using SPSS (27) shown in Table 3, the KMO value was 0.967, Bartlett’s spherical test chi-square value was 11,647.355, and the significance level was significant (*p* < 0.001). It means the data quality was significant, and the dimensions were good enough for the reliability analysis. At the same time, the tests of the factor loading matrix and the variance contribution ratio were all consistent with and above the threshold value. It meant the model construct data were scientifically valid and had good structural validity.

Principal component analysis was used to pull out the factors, as shown in Table 4. The cumulative variance contribution rate reached 76.662%, reflecting the original data better. The orthogonal rotation method used to find the common factors, the factor composition, and the model’s proposed hypothesis all match up, which shows that the data have good structural validity.

### 4.2. Validity Test between Latent Variables

The validity and peak values of every variable in the model presented in this work satisfy the specifications, proving that the distribution of variables satisfies the multivariate normality assumption. The updated model factor loadings all fulfil the requirements after looking at the external model loadings of the observed variables, excluding the observed variables with values below 0.7 and excluding Q1, Q13, Q16, Q18, Q22, Q25, and Q28 observed variables.

The latent model variable (Cronbach alpha) values and the composite reliability obtained by SmartPLS (version 3.0) analysis were between 0.813 and 0.928 and above the standard value of 0.7, respectively. The structural and differential validity included in the validity analysis might represent the consistency between the constructs and the validity between the measured variables. Table 5 shows that convergent validity can be inferred from composite reliability scores (CR) above 0.70. The (AVE) statistic was also taken out to evaluate convergent validity. The average variance indicated a suitable level of convergent validity (AVE) recovered for all structures in this analysis, more significant than 0.5 [51]. At the same time, the latent factors explained more than half of the variance in the indicators. According to Fornell and Larcker in 1981, the bottom triangle and the diagonal AVE open-root values in the table represent the Pearson correlations of dimensions [52]. All the article constructs were found to have open-root values higher than the other correlation factors, indicating that they had discriminant validity. The factor loadings of each indicator are examined as a second method of assessing the discriminant validity; the indicator should load more firmly on the target construct than any other variable [53]. The factor loading values for each construct (color labeling) were more significant than the other cross-factor loadings, as seen in Table 6. The executive summary suggests that the model has sufficient convergent and discriminant validity. The findings show that the measures utilized in this study are connected, the test items are reliable, and the cognitive response scores show internal consistency.

### 4.3. Homogeneous Variance Test

The artificial correlation between the explanatory and outcome variables induced by the same data source, measurement setting, survey context, or survey-specific characteristics is known as standard method variance (CMV). In this study, we first used the Harman one-way test to assess the homogenous variance. We discovered a single factor’s non-rotational maximum cumulative variance contribution (46.313%) was less than 50%. There was no issue with homogenous variance because all components combined contributed 76.662%. The model was then subjected to a Full Collinearity Assessment (FCA). When Variance Inflation Factor (VIF) values were less than 3.3, it was determined that the model did not have a homogenous variance problem. Thus, by combining the two tests mentioned above, the homoscedasticity issue has little impact on this study.

### 4.4. Model Fit Test

In order to evaluate the structural model in PLS-SEM, this paper first used the standardized root mean square of residuals (SRMR) of the model, and the results showed the SRMR = 0.071 (less than 0.1 is the passing level). The results were then based on the results suggested in the literature by Hair et al., such as the covariance (VIF) problem, the variance interpretation of the model (R^2^), the incremental interpretation of the exogenous variables on the endogenous variables (ƒ^2^), and the model predictive correlation (Q^2^). The VIF values for all constructs were evaluated below the threshold value of 3.3 [47], and the results did not show multicollinearity problems. ƒ^2^ effect size values showed that all relationships showed substantial effects. It is because values of 0.02, 0.15, and 0.35 represent small, medium, and significant effects of exogenous latent variables, respectively, according to the guidelines for assessing ƒ^2^ [48].

The model was then analyzed, and the R^2^ values ranged from 0.53 to 0.861. The constructs are within the ideal range for model interpretation. ƒ^2^ has a value between 0.026 and 0.473, which meets the threshold requirement. Predictive relevance was assessed using the Blindfolding algorithm procedure, and the distance seven was chosen to be omitted to ensure that the observed value (453) in the model of this study was not an integer divided by 7. Therefore, the Q^2^ values are more significant than 0 between 0.309 and 0.669, indicating that the model has sufficient predictive relevance. The significant path analysis of each variable was first tested by running 5000 times using Bootstrapping, as shown in Table 7. The path was refined to obtain the modified model, as Figure 2 shows path significance analysis for each variable.

The results of the structural equation correction model (Figure 2) show that (H1) public support for environmental protection policies has a considerable positive impact on the foundation of public engagement in environmental participation (β = 0.640, *p* < 0.001). According to hypothesis (H2a,b), there is a tenuous positive correlation between popular support for environmental protection policies and public awareness and competency in environmental protection (β = 0.729, *p* < 0.001; β = 0.500, *p* < 0.001). As can be seen, both types of public participation in environmental protection subjects and consequences benefit significantly from and are nurtured by public support for environmental protection policy.

As public participation in environmental protection awareness has a positive impact on the participation base of public participation in environmental protection, with an influence coefficient for the dimension of public participation subject’s awareness and ability, the third hypothesis (H3a) is proposed in this paper (β = 188, *p* < 0.001). The ability of the people to participate in environmental protection on a participation basis path hypothesis (H3b) is false and has no appreciable impact. The ability of the public to contribute to environmental protection is significantly impacted by public awareness of environmental protection (β = 0.405, *p* < 0.001) (H4).

On the analysis of information disclosure and environmental satisfaction, the fifth hypothesis (H5a,b) proposed in this paper, public participation in environmental protection policy support has a significant positive effect on the actual function of public participation in environmental protection performance (β = 0.437, *p* < 0.001) (H5a), as well as a significant positive effect on public participation in environmental protection performance satisfaction (β = 0.201, *p* < 0.001) (H5b).

Meanwhile, this paper’s sixth hypothesis (H6a,b) is that public participation in environmental protection awareness has no significant effect on public participation. The public participation in environmental protection capacity has a positive effect on both public participation in environmental protection participation basis and public participation in environmental protection performance actual function (H7a,b); the path coefficients are (β = 0.114, *p* < 0.05) (H7a) and (β = 0.174, *p* < 0.01) (H7b), respectively. Public participation in environmental protection performance satisfaction (H8a-b) is significant; the coefficient of effect is, respectively (β = 0.293, *p* < 0.001) (H8a); (β = 0.576, *p* < 0.001) (H8b).

Overall, all assumptions are accurate except for H3b and H6a,b. The central claim of this article is that public support for environmental protection policies positively impacts how people behave. Most government policies emphasize the advancement of public participation in environmental protection knowledge and the promotion of public participation in environmental protection capacity as having a positive direction. The national policy’s emphasis on building a beautiful China is particularly underlined by the consciousness orientation, which also affects consciousness behaviour and drives behaviour growth. By enhancing the actual function of public participation in environmental protection information performance and realizing the comprehensive development among all dimensions in constructing the result-oriented mechanism, the consciousness orientation also enhances the effect of public participation in environmental protection.

### 4.5. Indirect Effect Analysis

The main indirect impacts of the structural equation model are listed in Table 8. We can determine the relative strength of each path by examining the percentage of the specific indirect effect in the total indirect effect. We can determine the relative potency of each pathway by examining the specific indirect effect as a percentage of the total indirect effect. The results demonstrate a few columns of essential pathways concerning the mediating role of the subjective quality and participation base between policy support and policy success in the integrated model, in line with expectations. Policy support shows significant indirect effects on both the actual function of policy performance (β = 0.392, *p* < 0.001) and performance satisfaction (β = 0.633, *p* < 0.001).

Further analysis of the specific indirect effects revealed that the path of policy support → public participation base (β = 0.199, *p* < 0.001; β = 0.390, *p* < 0.001) explained the vast majority of the total profile effect between both policy support and policy performance. They indicate that the upland pair of public participation base is the most critical mediating variable in policy support acting on policy performance. The indirect effects shown by the path via subjective quality are minimal.

## 5. Discussion

The impact of public policy support on public policy performance in the context of creating a beautiful China is examined in this study using a web-based questionnaire. On the one hand, it incorporates subjectivity into the model and conceptually distinguishes the foundation for involvement in public policy from the execution of public policy. There is a dismissal of subjectivity and a mention of the “double processing” theory. It highlights the division of subjectivity into public participation and how the interaction of emotion and reason results in the feeling of risk. Public involvement awareness and public participation capability are the two kinds of subjective components. Comparatively to the causal chain model, the integrated model proposed in this study gives a thorough picture of the complicated and quick paths of policy support on policy performance. In order to escape the Tacitus trap, relevant government entities must direct policy creation toward contributing to policy performance. The outcomes are essential for understanding policy inputs, such as policy efficacy following policy support.

The results of structural equation modelling revealed that there is a significant process impact of government policy input on policy performance: the more policy input, the better the overall improvement in policy performance, and the extensive level of public participation in environmental protection is enhanced through the system of policy participation base formation, which also tends to have a driving effect on government trust, as a logical process, which can simplify the complexity of public behaviour by offering the essential guiding principles for social engagement. As a result, improving government credibility and execution and establishing a people-focused, service-oriented government are crucial for developing successful policies. The government continues to play a significant role in raising public involvement in environmental protection. It identifies the requirements for effective policy effectiveness.

However, it is also a subjective reaction to the regulator’s poor oversight, insufficiency, or even absence if the consequence of policy implementation during policy formation lowers the government’s confidence. It is due to public involvement in environmental protection and the legitimacy of the government as a foundation for engagement. The public will inevitably distrust the government. The confidence issue also makes it easier for the government to accelerate its efforts to build credit and carefully evaluate the effects of risky policies. It is imperative to increase public feedback, improve the efficiency of government operations, and decrease public mistrust to stem the long-term, deepening erosion of public confidence in the legitimacy of government governance. When implementing programs to enhance public expectations and calm people’s hearts, avoiding slipping into the Tacitus trap is crucial.

Government legitimacy is, therefore, the most important mediating factor for the implementation effect after policy input. Second, the participation base demonstrates how indirect governmental policy success and support for such programs interact. More public trust in the government may lead to more accessible risk communication, but this should never be the primary goal of public policy measures. The general public is a rational group that bases its decisions on the truth and engages with its surroundings to reach logical conclusions. It is not a group that absorbs information without question. In order to reduce public uncertainty during the consultation and interaction process, relevant government departments must give up any sense of superiority when developing policies, provide the public with open and transparent information for logical evaluation, and create a platform for reciprocal communication with the public. It will encourage emotional identification and prevent scenarios where much unfavorable information leads to poor public engagement. The general public must believe that the government communicates constructively and transparently. When necessary, it must take the initiative to assume accountability and swiftly, correctly, and impartially inform the public of legislative or executive action. On the other hand, it needs to understand and accept the needs of the general public. However, it can provide a comprehensive understanding of people’s responses, allowing for the enhancement and further modification of risk response strategies.

An in-depth analysis of the analysis reveals that the “central persuasion path” is challenging to follow and that the public’s capacity and desire to evaluate outside information in public participation in environmental protection are limited by their specialized knowledge and information sources. The “pivotal persuasion path”, which entails carefully processing detailed information, thinking, analyzing, and summarizing the characteristics of public participation in environmental protection—the ratio of benefits and developing attitudes and behavioral intentions for public participation—presents challenges to the general public. The public frequently choose the “marginal persuasion path” instead. The public are more inclined to follow the “marginal persuasion path”, which connects the risk assessment of domestic immunizations to other cues such as trust as a source of risk counselling, even when faith in the government is not always reasonable. The government must balance the immediate consequences and long-term viability of risk management and communication strategies based on differentiating the connotations of various public engagement events and enlisting technical experts to ensure information accuracy and scientific communication. However, compared to the main path, the marginal path’s capacity to forecast behavioral intentions and the permanence of attitude changes it causes are significantly lower.

Additionally, it should provide a two-way decision-making path for the general public under specific hazards and establish a long-lasting dialogue and feedback system to meet their cognitive ability and psychological needs. It will also be tested further in the following experiments to determine how the degree of desire connected to different behavioral intention factors will affect the analysis outcomes of the integrated model. Such behavioral intentions only apply to respondents, their family members, and close associates, as indicated by respondents’ desire to adopt them actively.

## 6. Conclusions

### 6.1. Research Conclusions

By undertaking this study, we aim to increase the environmental norms for how involvement affects governance outputs, the arguments for their acceptance, and the intermediate results that serve as modifiers of this relationship. The involvement results often favor environmental governance standards, particularly when participants are given significant control over decisions and outcomes, according to the empirical findings of our case-finding meta-analysis of published case studies. However, it was shown that components of intensive communication had a considerable impact on early social and cooperative outcomes. Notably, only one of these two intermediary social outcomes—the “convergence of stakeholder perspectives”—had a perceptible influence on environmental output standards and a more substantial influence on acceptability, indirectly influencing the implementation of environmental output standards. This element covers interpersonal aspects of conflict resolution, the development of trust, mutual gain, and the creation of norms. While “stakeholder capacity building” (which encompasses components such as social learning, personal capacity building, and network creation) was determined to be important in and of itself, it was not discovered to impact environmental outcomes substantially. This analysis illustrates the perspective obtained by carefully analyzing complex and multifaceted circumstances more generally. It examines how their various aspects might promote social and cooperative outcomes and perhaps enhance the environmental effects of public decision-making procedures. It paves the path for further study in several fields.

The interaction of general awareness, social conditions, and cognitive preferences influences public participation in environmental conservation. It is further noted that the effect is evident due to the beautiful China construction’s dominant and directing position in the general consciousness. Future policy development in connected sectors should fully consider the influence of social factors and cognitive preferences to prevent policy failure. First, it is essential to improve policy research. Before putting the policy into force, a thorough analysis should be conducted to ascertain its goals, content, implementation method, efficacy, and viability to choose the optimal policy implementation strategy. Improve the capacity for implementing policies next. In order to improve the efficiency of policy implementation and reduce errors in policy implementation, skill training for policymakers and implementers should be improved to increase the effectiveness of policy implementation and decrease policy implementation errors. Third, improve policy oversight. A suitable policy oversight mechanism should be set up to detect policy implementation issues and conduct prompt and efficient policy adjustments to ensure effective policy implementation. Strengthening policy evaluation is the fourth step. An effective policy evaluation mechanism should be established to ensure effective policy implementation. This system should be used to evaluate the results of policy implementation routinely, quickly spot problems with policy implementation, and swiftly put effective remedial measures into place. Finally, improve the communication of policy: Communication between policy implementers and policy beneficiaries needs to be enhanced to encourage efficient policy implementation. It will increase policy beneficiaries’ understanding of the policy and lessen resistance to its implementation.

Enhance the performance work’s effect orientation while focusing on the performance’s fundamentals and policy orientation. To assure the normality of the appraisal, create a faultless performance appraisal system, specify the scope, substance, techniques, criteria, and results, and strictly enforce them. Clarify the accountable party, participant, process, timing, and incentives of the appraisal to ensure the efficacy of the appraisal. It will help you create the ideal performance appraisal mechanism. The outcomes of the appraisal should accurately reflect the employee’s actual performance in order to ensure the success of the process and the appraisal’s objectivity. This will give employees immediate feedback on the evaluation results to remedy their errors, improve their job, and increase efficiency. The policy’s inverted “U” impact is reduced to avoid the “thunder but little rain” scenario.

### 6.2. Implications, Limitations, and Future Research

This research examines public involvement in environmental protection, which raises the environmental protection assessment method and improves the assessment ratio while also exploring the assessment effect under the influence of various elements. Providing more effective motivational processes and strategies is crucial to reach the ultimate aim of environmental preservation. These strategies must also better promote public participation in environmental protection. However, we also discovered certain flaws. Data from multiple institutional and environmental contexts were used in this investigation.

Further research may clarify these contextual aspects to understand better “what works when and how” in participatory and collaborative governance. Unexpected findings from this study included the direct impact of delegated authority on output environmental requirements. A follow-up study may better examine this relationship to comprehend the mechanisms at play in this circumstance and produce new concepts about these links. This study could grow if more outcome categories were included. Our approach produced the worst outcomes when it was put into practice. Future studies must tackle the challenge of adequately describing the implementation of and compliance with agreed results on the ground and the role of collaborative and participatory governance procedures. The first step in this direction might be to supplement our study with other data collecting, including media analysis or interviews, to provide a complete examination of implementation.

## Figures and Tables

**Figure 1 ijerph-20-05084-f001:**
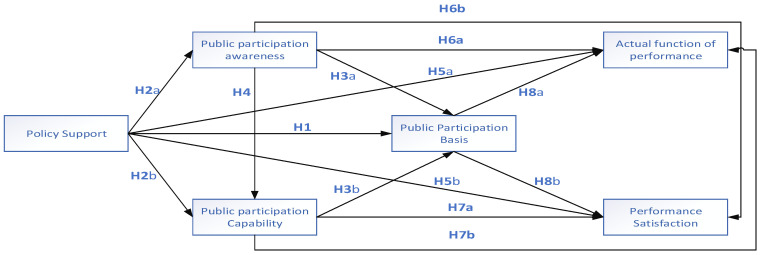
Public participation in the environmental protection model.

**Figure 2 ijerph-20-05084-f002:**
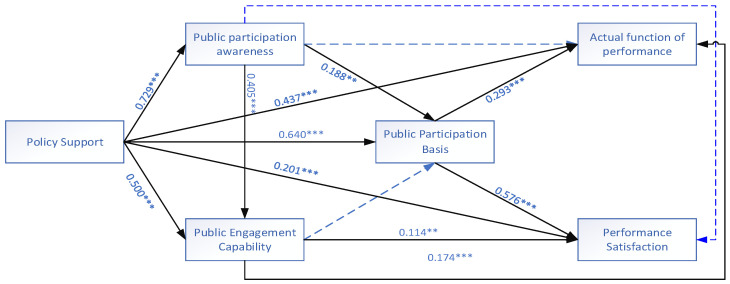
Final model of public participation in environmental protection. Note: (1) Presented as standardized coefficients in the model; (2) two-tailed statistical test levels: ** *p* < 0.01, *** *p* < 0.001 (*n* = 453).

**Table 1 ijerph-20-05084-t001:** Provides a summary of the respondents’ (*n* = 453) demographic information.

Item	Category	*n* (%)	Item	Category	*n* (%)
Sex	Male	221 (48.8)	Education Level	Medium Specialized	19 (4.2)
	Female	232 (51.2)		University Undergraduate	226 (49.9)
Age	18~23	24 (5.3)		Postgraduate and above	171 (37.7)
	24~30	237 (50.1)	Family income	Under RMB 4000	46 (10.2)
	31~40	126 (27.8)		RMB 4000 to 8000	138 (30.5)
	Over 40 years old	66 (16.8)		RMB 8000 to 12,000	151 (33.3)
Education Level	High school, junior college or below	37 (8.2)		Over RMB 12,000 RMB	118 (26)

**Table 2 ijerph-20-05084-t002:** Measurement items and descriptive statistics of variables.

Variable	Measurement Items	Code	Mean	SD
Policy Support	The government has put in place a lot of rules and policies to help protect the environment.	PS1	5.71	1.572
	I have participated in activities like environmental administrative permits and environmental administrative penalties that help protect the environment.	PS2	4.83	1.755
	I have participated in questionnaires, random interviews, symposiums, and online polls about protecting the environment.	PS3	4.68	1.783
	Public participation involves a lot of different kinds of people.	PS4	4.57	1.789
Public participation awareness	The economy is more important to our country than taking care of the environment.	PA1	3.51	2.050
	I see a lot of news, propaganda, and public service announcements about how the public can help protect the environment on TV and the Internet.	PA2	5.69	1.668
	I am very willing to participate in activities that protect the environment and help people.	PA3	5.98	1.545
	I am involved in many activities that help protect the environment.	PA4	4.44	1.908
	The government often sets up ways for the public to take part in environmental assessment and big decisions about the environment.	PA5	4.77	1.957
	I will do everything possible to stop things that hurt the environment.	PA6	5.01	1.814
Public participation capability	I have taken part in many pieces of training for protecting the environment.	PC1	3.66	2.002
	More and more groups are working for the public good to protect the environment.	PC2	5.19	1.76
	Many schools now have classes that teach how to protect the environment.	PC3	5.06	1.844
	I have a professional understanding of policies, laws, and rules for protecting the environment.	PC4	4.69	1.674
Public participation basis	Environmental protection has gotten better over the past few years.	PB1	5.21	1.622
	The variety is the publicity methods to promote public participation in the environmental protection atmosphere.	PB3	5.16	1.678
	Having the public help protect the environment has a good effect.	PB4	5.71	1.547
	People helped protect the environment by making comments and suggestions, and the local government responded promptly and helpfully.	PB5	4.87	1.738
	The public needs to help solve pollution problems better.	PB6	3.85	2.043
Actual function of performance	Local government websites are full of information about protecting the environment.	FP1	4.96	1.734
	The public finds it easy to accept and understand information about environmental protection from their local government.	FP2	4.90	1.719
	Many companies talk about their efforts to protect the environment.	FP3	4.50	1.802
	I know many unique websites and WeChat numbers for government and business environmental protection information disclosure.	FP4	4.47	1.836
	I know how to ask for information about environmental protection.	FP5	4.44	1.909
Performance satisfaction	I know that the government has passed rules to make it easier for the public to help protect the environment.	FS1	4.85	1.839
	The government still gives environmental groups much help.	FS2	4.86	1.727
	The government is improving at helping the public participate in policies that protect the environment.	FS3	5.00	1.790
	There are more and more ways for the general public to help protect the environment.	FS4	5.27	1.636

Note: SD = standard deviation.

**Table 3 ijerph-20-05084-t003:** Evaluation index test results. Source: calculated and summarized by the author.

KMO Laboratory Value		0.967
Bartlett sphericity test	Approximate Chi-square	11,647.355
	df	378
	*p*-value	***

*** *p* < 0.001.

**Table 4 ijerph-20-05084-t004:** Total variance explained.

Ingredients	Initial Eigenvalue			Extraction of the Sum of Squares of Loads			The Sum of Squared Rotating Loads		
	Total	Percentage of Variance (%)	Accumulation (%)	Total	Percentage of Variance (%)	Accumulation (%)	Total	Percentage of Variance (%)	Accumulation (%)
1	14.768	46.313	46.313	14.768	46.313	46.313	7.828	27.958	27.958
2	1.746	16.235	62.548	1.746	16.235	62.548	4.212	15.044	43.002
3	1.387	4.955	67.503	1.387	4.955	67.503	3.82	13.644	56.646
4	1.076	3.844	71.347	1.076	3.844	71.347	3.451	12.326	68.973
5	0.792	2.827	74.174	0.792	2.827	74.174	1.096	3.916	72.888
6	0.697	2.488	76.662	0.697	2.488	76.662	1.057	3.774	76.662
7	0.608	2.172	78.834						
8	0.553	1.976	80.81						
9	0.522	1.864	82.674						
10	0.455	1.626	84.301						
11	0.443	1.582	85.883						
12	0.39	1.392	87.275						
13	0.363	1.297	88.572						
14	0.346	1.236	89.808						
15	0.326	1.163	90.971						
16	0.296	1.057	92.028						
17	0.282	1.006	93.034						
18	0.257	0.92	93.954						
19	0.242	0.863	94.817						
20	0.231	0.824	95.641						
21	0.216	0.77	96.411						
22	0.191	0.684	97.095						
23	0.169	0.604	97.699						
24	0.16	0.572	98.271						
25	0.145	0.519	98.791						
26	0.123	0.439	99.23						
27	0.114	0.406	99.636						
28	0.102	0.364	100						

Extraction method: principal component analysis.

**Table 5 ijerph-20-05084-t005:** Descriptive statistics and correlation coefficients of each variable.

Variables	Descriptive Statistics					Correlation Coefficient					
	Mean	SD	CA	CR	AVE	Public Participation Basis	Public Participation Awareness	Policy Support	Public Engagement Capability	Actual Function of Performance	Performance Satisfaction
1	5.261	1.634	0.868	0.919	0.791	**0.889**					
2	5.175	1.776	0.828	0.879	0.593	0.705	**0.770**				
3	5.035	1.703	0.813	0.890	0.729	0.829	0.729	**0.854**			
4	4.649	1.818	0.840	0.893	0.676	0.719	0.769	0.795	**0.822**		
5	4.632	1.813	0.883	0.928	0.810	0.791	0.673	0.828	0.743	**0.900**	
6	4.996	1.732	0.866	0.918	0.789	0.882	0.708	0.834	0.767	0.868	**0.888**

Note: (1) The square of the mean variance extracted value at the bolded content; (2) Mean = mean; AVE = mean variance extracted value; CA = Cronbach’s alpha coefficient; CR = combined reliability; SD = standard deviation; (3) the diagonal outline of the word is the AVE open-root value, and the lower triangle is the Pearson correlation of dimensionality.

**Table 6 ijerph-20-05084-t006:** Measurement model cross-loadings.

Observation Variables	Public Participation Basis	Public Participation Awareness	Policy Support	Public Engagement Capability	Actual Function of Performance	Performance Satisfaction
Q9	0.649	0.642	0.721	**0.851**	0.632	0.668
Q8	0.614	0.657	0.645	**0.834**	0.581	0.641
Q7	0.462	0.597	0.531	**0.780**	0.566	0.541
Q6	0.485	**0.774**	0.560	0.642	0.525	0.527
Q5	0.604	**0.817**	0.596	0.679	0.622	0.620
Q4	0.520	**0.794**	0.576	0.642	0.565	0.549
Q3	0.562	**0.760**	0.558	0.502	0.444	0.538
Q27	**0.895**	0.634	0.771	0.680	0.809	0.811
Q26	**0.843**	0.585	0.652	0.549	0.543	0.680
Q24	0.867	0.659	0.750	0.680	0.715	**0.875**
Q23	**0.929**	0.660	0.779	0.677	0.730	0.847
Q21	0.734	0.585	0.715	0.669	0.770	**0.888**
Q20	0.744	0.640	0.754	0.694	0.829	**0.901**
Q2	0.544	**0.697**	0.513	0.471	0.410	0.480
Q19	0.716	0.618	0.730	0.665	**0.902**	0.807
Q17	0.665	0.568	0.728	0.664	**0.902**	0.742
Q15	0.750	0.629	0.777	0.677	**0.897**	0.794
Q14	0.717	0.625	**0.877**	0.710	0.788	0.756
Q12	0.680	0.602	**0.875**	0.690	0.713	0.699
Q11	0.623	0.634	0.699	**0.823**	0.659	0.661
Q10	0.729	0.642	**0.807**	0.635	0.614	0.677

**Table 7 ijerph-20-05084-t007:** Path significance analysis for each variable.

Latent Variable Pathway Correlation	T-Value	*p*-Value	Significant
Public Participation Basis → Actual function of performance	5.211	***	Yes
Public Participation Basis → Performance Satisfaction	13.357	***	Yes
Public participation awareness → Public Participation Basis	3.497	***	Yes
Public participation awareness → Public Engagement Capability	9.19	***	Yes
Public participation awareness → Actual function of performance	0.246	0.806	No
Public participation awareness → Performance Satisfaction	0.44	0.66	No
Public Participation Basis → Public Participation Basis	14.185	***	Yes
Public Participation Basis → Public participation awareness	27.48	***	Yes
Public Participation Basis → Public Engagement Capability	11.427	***	Yes
Public Participation Basis → Actual function of performance	7.207	***	Yes
Public Participation Basis → Performance Satisfaction	3.562	***	Yes
Public Engagement Capability → Public Participation Basis	1.105	0.27	No
Public Engagement Capability → Actual function of performance	3.025	**	Yes
Public Engagement Capability → Performance Satisfaction	3.554	***	Yes

Two-tailed statistical test levels: ** *p* < 0.01, *** *p* < 0.001 (*n* = 453).

**Table 8 ijerph-20-05084-t008:** Summary of indirect effects.

		β-Coefficient	T-Value	*p*-Value
Policy Support to Performance Actual Function	Policy Support → Public Engagement Capability → Actual function of performance	0.091	3.340	0.001
	Policy Support → Public Participation Basis → Actual function of performance	0.199	5.436	<0.001
	Policy Support → Public participation awareness → Public Engagement Capability → Actual function of performance	0.054	3.314	0.001
	Policy Support → Public participation awareness → Public Participation Basis → Actual function of performance	0.046	3.552	<0.001
	Total indirect effect	0.392	8.135	<0.001
Policy Support to Performance Satisfaction	Policy Support → Public Engagement Capability → Performance Satisfaction	0.094	3.946	<0.001
	Policy Support → Public Participation Basis → Performance Satisfaction	0.390	11.071	<0.001
	Policy Support → Public participation awareness → Public Engagement Capability → Performance Satisfaction	0.055	3.601	<0.001
	Policy Support → Public participation awareness → Public Participation Basis → Performance Satisfaction	0.091	4.138	<0.001
	Total indirect effect	0.633	13.170	<0.001

Note: Only significant effects are summarized here; the sum of the specific effects does not equal the total indirect effect.

## Data Availability

All the data will be available from the corresponding author after reasonable request.

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
