# Peer review of "Research on Motivational Mechanisms and Pathways for Promoting Public Participation in Environmental Protection Behavior"

_ijerph, 2023, doi:10.3390/ijerph20065084_

Round 1

Reviewer 1 Report

The concept of Beautiful China needs to be better explained in the abstract.  Most of the sentences, especially in the introduction are complex, long, and have multiple concepts included, which may be revised with simple and straight sentences so that the readability is improved.

The methodology covers multiple hypotheses H1 to H8 with a & b subsections. It will be better if the results are also presented in the same logical sequence. Some part under the section is presented in future tense which needs correction.

Based on the results a concise build-up of a new public engagement theory can be attempted which is found missing. This would have been a better contribution to the theoretical base of environment protection policy research.

Author Response

We re-generalized and categorized the idea of Beautiful China in the introduction. Sentence breaks and the avoidance of lengthy, complex sentences were used in the revision of this section. The article's concept, goal, and scientific questions under investigation have all been further elucidated, and its readability and relevancy have increased.By rewriting the analysis section, we discuss the conceptual framework and findings from H1 to H8 for comparative analysis and rewrite them in the article's proper order. The article's reasoning is now more substantial after editing, and it is now easier to read.The relevance of the paper's theoretical contribution is increased by the further explanation of the connotation of public involvement in environmental preservation that we provide in the conclusion section.

    A:Our summary section was modified to read:Public participation in environmental protection is an essential component of evaluating the effectiveness of ecological and environmental protection. General awareness, social dynamics, and cognitive preferences frequently impact protection's impact. To investigate the correlation research of the confluence of mainstream awareness, social factors, and cognitive preferences by building a theoretical model. First, this work employs partial least squares structural equation modelling (PLS-SEM). Second, using the mediation model, the research describes and examines the factors that motivate public involvement in ecological and environmental conservation. Third, the research summarizes the suggested path countermeasures to offer practical advice and helpful ecological and environmental protection solutions. The findings demonstrate that mainstream policy leadership substantially impacts environmental conservation. Leadership in policy matters restricts the group's natural awareness of social factors. The subjective quality and competence basis in cognitive preferences are significantly influenced by policy leadership. Policy leadership significantly influences the effectiveness of environmental protection through the mediating factor of cognitive preferences. The ability base has a considerable mediating effect on cognitive preferences.

B:We present in the article in a hypothetical order:The results of the structural equation correction model (Figure 2) show that (H1)public support for environmental protection policies has a considerable positive impact on the foundation of public engagement in environmental participation(β=0.640,p<0.001). According to hypothesis H2(a-b), there is a tenuous positive correlation between popular support for environmental protection policies and public awareness and competency in environmental protection(β=0.729,p<0.001;β=0.500,p<0.001). As can be seen, both types of public participation in environmental protection subjects and consequences benefit significantly from and are nurtured by public support for environmental protection policy.

Although public participation in environmental protection awareness has a positive impact on the participation base of public participation in environmental protection, with an influence coefficient of for the dimension of public participation subject's awareness and ability, the third hypothesis (H3a) is proposed in this paper(β=188,p<0.001). The ability of the people to participate in environmental protection on a participation basis path hypothesis (H3b) is false and has no appreciable impact. The ability of the public to contribute to environmental protection is significantly and significantly impacted by public awareness of environmental protection(β=0.405, p<0.001)(H4).

On the analysis of information disclosure and environmental satisfaction, the fifth hypothesis (H5a-b) proposed in this paper, public participation in environmental protection policy support has a significant positive effect on the actual function of public participation in environmental protection performance (β=0.437, p<0.001)(H5a), as well as a significant positive effect on public participation in environmental protection performance satisfaction (β=0.201, p<0.001)(H5b).

Meanwhile, this paper's sixth hypothesis (H6a-b) is that public participation in environmental protection awareness has no significant effect on both public participation. For public participation in environmental protection capacity has a positive effect on both public participation in environmental protection participation basis and public participation in environmental protection performance actual function (H7a-b), the path coefficients are (β=0.114, p<0.05)(H7a) and (β=0.174, p<0.01)(H7b) respectively. public participation in environmental protection performance satisfaction (H8a-b) is significant, the coefficient of effect respectively (β=0.293, p<0.001)(H8a); (β=0.576, p<0.001)(H8b).

C:We have provided a further overview and distillation of public participation in environmental protection:The junction of general awareness, social conditions, and cognitive preferences influences public participation in environmental conservation. It is further noted that the effect is evident due to the Beautiful China construction's dominant and directing position in the general consciousness. Future policy development in connected sectors should fully consider the influence of social factors and cognitive preferences to prevent policy failure. First, it is essential to improve policy research. Before putting the policy into force, a thorough analysis should be done to ascertain its goals, content, implementation method, efficacy, and viability to choose the optimal policy implementation strategy. Improve the capacity for implementing policies next. In order to improve the efficiency of policy implementation and reduce errors in policy implementation, skill training for policymakers and implementers should be improved to increase the effectiveness of policy implementation and decrease policy implementation errors. Third, improve policy oversight. A suitable policy oversight mechanism should be set up to detect policy implementation issues and conduct prompt and efficient policy adjustments to ensure effective policy implementation. Strengthening policy evaluation is the fourth step. An effective policy evaluation mechanism should be established to ensure effective policy implementation. This system should be used to evaluate the results of policy implementation routinely, quickly spot problems with policy implementation, and swiftly put effective remedial measures into place. Finally, improve the communication of policy: Communication between policy implementers and policy beneficiaries needs to be enhanced to encourage efficient policy implementation. It will increase policy beneficiaries' understanding of the policy and lessen resistance to its implementation.

Reviewer 2 Report

I think the authors have understood the research process well, but it is better to consider the following points to improve the work.

I think more results should be added to the abstract section

The introduction section is good, but I suggest adding the following items as more evidence to this section:

* Savari, M., & Khaleghi, B. (2023). Application of the extended theory of planned behavior in predicting the behavioral intentions of Iranian’s local communities toward forest conservation. Frontiers in Psychology14, 33.

It is better to explain research sampling in the materials and methods section

Be sure to submit the questionnaire in the materials and methods section

Provide the validity and reliability of the research tool.

In the results section, be sure to bring the status of the variables so that I can see how their status was in your area.

Presenting practical policies can contribute to the attractiveness of this research

Author Response

We revised the abstract again to refine the article's meaning and present more of its findings. We also read the article "Application of the extended theory of planned behavior in predicting the behavioral intentions of Iranian's. We also read the article "Application of the extended theory of planned behavior in predicting the behavioural intentions of Iranian's local communities toward forest conservation" and gained a lot from it, and used many of the ideas and theories in it as theoretical support for our article. Once again, I would like to thank the editors for their recommendation of the literature, which is excellent advice and recommendation for our team.We made adjustments in response to the ideas one by one, and we included the respondents' locations in the questionnaire. The whole questionnaire's themes and objectives are displayed in tabular format. In the materials and methods section, we describe the research sampling while also debating the reliability and validity of the research instrument in-depth. We provide a thorough analysis of the practical steps that the public may take to participate in environmental protection and offer workable solutions to help advance both environmental protection and ecological protection.

A:We provide a more comprehensive explanation of the relevant theory in the introduction:The central government has been stepping up the environmental examination of local governments, and environmental protection in China has recently taken on a high-pressure stance [1]. The public's engagement in performance evaluation is a crucial component of environmental protection assessment, a methodical effort including the government, businesses, and the general public. Enhancing the efficiency of public engagement in environmental protection performance assessment is one of them. By doing so, the adverse effects of public value conflicts on the effectiveness of environmental governance can be successfully mitigated [2].

A kind of "social environmentalism" is the public's involvement in environmental conservation. It supports public participation in environmental protection initiatives, sees environmental protection as a civil right, and sees the general public as its wards and participants [3]. The social environment is an intricate, dynamic system. According to social environmentalism, it is essential to consider how people's social environments affect their behavioral choices and thoughts, behaviors, and outcomes [4]. Encouraging public involvement in environmental protection can assist in increasing public knowledge and duty to participate in environmental protection activities and improve the performance outcomes of environmental protection evaluation. Although the social environment is objectively conducive to public involvement in environmental protection at this point [5], what elements influence the people's passion for participation, and what is most important among them?

Performance refers to a system or process's actions or outcomes [6]. Performance is a result, genuine behavior, or performance [5]. According to the theory of planned behavior, a model of behavioral explanation, organizational drivers have a significant role in determining human behavior, perceptions and attitudes, behavioral skills, environment, and behavior outcomes [7]. In light of the cascading environmental strain and condensing workload of the central government, what are the realistic ways to enhance public participation in environmental protection performance?

B:We have added a number of elements to the questionnaire section:Because collecting nationwide survey samples using precise probability sampling is challenging, this study used Questionnaire Star, a recognized online questionnaire survey platform. As the leading research site, we primarily chose the province of Qinghai, where there is good public involvement in environmental conservation. We conducted the questionnaire interviews at random locations while following the survey's requests. The authors employed the questionnaire research platform to carry out a similarity test and an invalid questionnaire filtering review process on the data collected to assure the data quality's reliability and accuracy. These methods resulted in the collection of 453 valid questionnaires for this study, of which 48.8% (n = 221) of the respondents were men, 51.2% (n = 232) were women, and 87.6% (n = 453) held a bachelor's degree or higher. The demographic characteristics of the respondents are detailed in Table 1.

Table 2. Measurement items and descriptive statistics of variables.

Variable

Measurement Items

Code

Mean

SD

Policy Support

The government has put in place a lot of rules and policies to help protect the environment.

PS1

5.71

1.572

I have participated in activities like environmental administrative permits and environmental administrative penalties that help protect the environment.

PS2

4.83

1.755

I have participated in questionnaires, random interviews, symposiums, and online polls about protecting the environment.

PS3

4.68

1.783

Public participation involves a lot of different kinds of people.

PS4

4.57

1.789

Public participation awareness

The economy is more important to our country than taking care of the environment.

PA1

3.51

2.050

I see a lot of news, propaganda, and public service announcements about how the public can help protect the environment on TV and the Internet.

PA2

5.69

1.668

I am very willing to participate in activities that protect the environment and help people.

PA3

5.98

1.545

I am involved in many activities that help protect the environment.

PA4

4.44

1.908

The government often sets up ways for the public to take part in environmental assessment and big decisions about the environment.

PA5

4.77

1.957

I will do everything possible to stop things that hurt the environment.

PA6

5.01

1.814

Public participation capability

I have taken part in many pieces of training for protecting the environment.

PC1

3.66

2.002

More and more groups are working for the public good to protect the environment.

PC2

5.19

1.76

Many schools now have classes that teach how to protect the environment.

PC3

5.06

1.844

I have a professional understanding of policies, laws, and rules for protecting the environment.

PC4

4.69

1.674

Public participation basis

Environmental protection has gotten better over the past few years.

PB1

5.21

1.622

The variety is the publicity methods to promote public participation in the environmental protection atmosphere.

PB3

5.16

1.678

Having the public help protect the environment has a good effect.

PB4

5.71

1.547

People helped protect the environment by making comments and suggestions, and the local government responded promptly and helpfully.

PB5

4.87

1.738

The public needs to help solve pollution problems better.

PB6

3.85

2.043

Actual function of performance

Local government websites are full of information about protecting the environment.

FP1

4.96

1.734

The public finds it easy to accept and understand information about environmental protection from their local government.

FP2

4.90

1.719

Many companies talk about their efforts to protect the environment.

FP3

4.50

1.802

I know many unique websites and WeChat numbers for government and business environmental protection information disclosure.

FP4

4.47

1.836

I know how to ask for information about environmental protection.

FP5

4.44

1.909

Performance satisfaction

I know that the government has passed rules to make it easier for the public to help protect the environment.

FS1

4.85

1.839

The government still gives environmental groups much help.

FS2

4.86

1.727

The government is improving at helping the public participate in policies that protect the environment.

FS3

5.00

1.790

There are more and more ways for the general public to help protect the environment.

FS4

5.27

1.636

Note: SD = standard deviation.

C:We have added the following modifications to the response:Enhance the performance work's effect orientation while focusing on the performance's fundamentals and policy orientation. To assure the normality of the appraisal, create a faultless performance appraisal system, specify the scope, substance, techniques, criteria, and results, and strictly enforce them. Clarify the accountable party, participant, process, timing, and incentives of the appraisal to ensure the efficacy of the appraisal. It will help you create the ideal performance appraisal mechanism. The outcomes of the appraisal should accurately reflect the employee's actual performance in order to ensure the success of the process and the appraisal's objectivity. To give employees immediate feedback on the evaluation results to remedy their errors, improve their job, and increase efficiency. The policy inverted "U" impact is reduced to avoid the "thunder but little rain" scenario.

Reviewer 3 Report

This article is relevant concerning the given topic, but there is a need to revise such places in the article:

·         Some sentences could be finalized (in Abstract and some places in text) – “Using a web-based questionnaire, empirical research of the motivational factors and influence pathways of the public's participation in environmental protection” – no clear idea from this sentence.

·         References in the text must be improved – now they are given not in correct way – e.g. <…construction(wang,2022).>

·         There is no clear research aim of the article – it should be mentioned.

·         Theory part could more structured – exploring concrete ideas then developing it, or dividing it in subsections.

·         Probably some concrete research examples concerning the topic could be mentioned on international debates?

·         In “3. Methodology” part nothing is written about questionnaire – it should be mentioned what type questions or question groups were included, what were their aims, etc.

·         What could be an example from this article for other articles – what things to rethink, to pay more attention, etc.?

Author Response

In exact agreement with the specifications for the piece, we changed the literature format. We added subheadings to the article to further improve its scientific rigour and classify it more scientifically. In addition, our team strengthened the article's rigour and scientific rigour by introducing examples from the worldwide literature. In order to better highlight the theoretical and practical contributions of the essay, we refine the practical path and experience summary before concluding with a discussion and summary that is more scientific and effective.Many of the contents have been reported earlier, so we will not repeat them.

 A:Our summary section was modified to read:Public participation in environmental protection is an essential component of evaluating the effectiveness of ecological and environmental protection. General awareness, social dynamics, and cognitive preferences frequently impact protection's impact. To investigate the correlation research of the confluence of mainstream awareness, social factors, and cognitive preferences by building a theoretical model. First, this work employs partial least squares structural equation modelling (PLS-SEM). Second, using the mediation model, the research describes and examines the factors that motivate public involvement in ecological and environmental conservation. Third, the research summarizes the suggested path countermeasures to offer practical advice and helpful ecological and environmental protection solutions. The findings demonstrate that mainstream policy leadership substantially impacts environmental conservation. Leadership in policy matters restricts the group's natural awareness of social factors. The subjective quality and competence basis in cognitive preferences are significantly influenced by policy leadership. Policy leadership significantly influences the effectiveness of environmental protection through the mediating factor of cognitive preferences. The ability base has a considerable mediating effect on cognitive preferences.

B:We have added a number of elements to the questionnaire section:Because collecting nationwide survey samples using precise probability sampling is challenging, this study used Questionnaire Star, a recognized online questionnaire survey platform. As the leading research site, we primarily chose the province of Qinghai, where there is good public involvement in environmental conservation. We conducted the questionnaire interviews at random locations while following the survey's requests. The authors employed the questionnaire research platform to carry out a similarity test and an invalid questionnaire filtering review process on the data collected to assure the data quality's reliability and accuracy. These methods resulted in the collection of 453 valid questionnaires for this study, of which 48.8% (n = 221) of the respondents were men, 51.2% (n = 232) were women, and 87.6% (n = 453) held a bachelor's degree or higher. The demographic characteristics of the respondents are detailed in Table 1.

Table 2. Measurement items and descriptive statistics of variables.

Variable

Measurement Items

Code

Mean

SD

Policy Support

The government has put in place a lot of rules and policies to help protect the environment.

PS1

5.71

1.572

I have participated in activities like environmental administrative permits and environmental administrative penalties that help protect the environment.

PS2

4.83

1.755

I have participated in questionnaires, random interviews, symposiums, and online polls about protecting the environment.

PS3

4.68

1.783

Public participation involves a lot of different kinds of people.

PS4

4.57

1.789

Public participation awareness

The economy is more important to our country than taking care of the environment.

PA1

3.51

2.050

I see a lot of news, propaganda, and public service announcements about how the public can help protect the environment on TV and the Internet.

PA2

5.69

1.668

I am very willing to participate in activities that protect the environment and help people.

PA3

5.98

1.545

I am involved in many activities that help protect the environment.

PA4

4.44

1.908

The government often sets up ways for the public to take part in environmental assessment and big decisions about the environment.

PA5

4.77

1.957

I will do everything possible to stop things that hurt the environment.

PA6

5.01

1.814

Public participation capability

I have taken part in many pieces of training for protecting the environment.

PC1

3.66

2.002

More and more groups are working for the public good to protect the environment.

PC2

5.19

1.76

Many schools now have classes that teach how to protect the environment.

PC3

5.06

1.844

I have a professional understanding of policies, laws, and rules for protecting the environment.

PC4

4.69

1.674

Public participation basis

Environmental protection has gotten better over the past few years.

PB1

5.21

1.622

The variety is the publicity methods to promote public participation in the environmental protection atmosphere.

PB3

5.16

1.678

Having the public help protect the environment has a good effect.

PB4

5.71

1.547

People helped protect the environment by making comments and suggestions, and the local government responded promptly and helpfully.

PB5

4.87

1.738

The public needs to help solve pollution problems better.

PB6

3.85

2.043

Actual function of performance

Local government websites are full of information about protecting the environment.

FP1

4.96

1.734

The public finds it easy to accept and understand information about environmental protection from their local government.

FP2

4.90

1.719

Many companies talk about their efforts to protect the environment.

FP3

4.50

1.802

I know many unique websites and WeChat numbers for government and business environmental protection information disclosure.

FP4

4.47

1.836

I know how to ask for information about environmental protection.

FP5

4.44

1.909

Performance satisfaction

I know that the government has passed rules to make it easier for the public to help protect the environment.

FS1

4.85

1.839

The government still gives environmental groups much help.

FS2

4.86

1.727

The government is improving at helping the public participate in policies that protect the environment.

FS3

5.00

1.790

There are more and more ways for the general public to help protect the environment.

FS4

5.27

1.636

Note: SD = standard deviation.

C:We have added the following modifications to the response:Enhance the performance work's effect orientation while focusing on the performance's fundamentals and policy orientation. To assure the normality of the appraisal, create a faultless performance appraisal system, specify the scope, substance, techniques, criteria, and results, and strictly enforce them. Clarify the accountable party, participant, process, timing, and incentives of the appraisal to ensure the efficacy of the appraisal. It will help you create the ideal performance appraisal mechanism. The outcomes of the appraisal should accurately reflect the employee's actual performance in order to ensure the success of the process and the appraisal's objectivity. To give employees immediate feedback on the evaluation results to remedy their errors, improve their job, and increase efficiency. The policy inverted "U" impact is reduced to avoid the "thunder but little rain" scenario.

Round 2

Reviewer 2 Report

the paper is good job